# Standardized Extract of *Asparagus officinalis* Stem Attenuates SARS-CoV-2 Spike Protein-Induced IL-6 and IL-1β Production by Suppressing p44/42 MAPK and Akt Phosphorylation in Murine Primary Macrophages

**DOI:** 10.3390/molecules26206189

**Published:** 2021-10-14

**Authors:** Ken Shirato, Jun Takanari, Takako Kizaki

**Affiliations:** 1Department of Molecular Predictive Medicine and Sport Science, Kyorin University School of Medicine, 6-20-2 Shinkawa, Mitaka, Tokyo 181-8611, Japan; kizaki@ks.kyorin-u.ac.jp; 2Amino Up Co., Ltd., 363-32 Shin-ei, Kiyota, Sapporo 004-0839, Japan; takanari@aminoup.jp

**Keywords:** *Asparagus officinalis* L., SARS-CoV-2, spike protein, inflammation, cell signaling, macrophage

## Abstract

Excessive host inflammation following infection with severe acute respiratory syndrome coronavirus 2 (SARS-CoV-2) is associated with severity and mortality in coronavirus disease 2019 (COVID-19). We recently reported that the SARS-CoV-2 spike protein S1 subunit (S1) induces pro-inflammatory responses by activating toll-like receptor 4 (TLR4) signaling in macrophages. A standardized extract of *Asparagus officinalis* stem (EAS) is a unique functional food that elicits anti-photoaging effects by suppressing pro-inflammatory signaling in hydrogen peroxide and ultraviolet B-exposed skin fibroblasts. To elucidate its potential in preventing excessive inflammation in COVID-19, we examined the effects of EAS on pro-inflammatory responses in S1-stimulated macrophages. Murine peritoneal exudate macrophages were co-treated with EAS and S1. Concentrations and mRNA levels of pro-inflammatory cytokines were assessed using enzyme-linked immunosorbent assay and reverse transcription and real-time polymerase chain reaction, respectively. Expression and phosphorylation levels of signaling proteins were analyzed using western blotting and fluorescence immunomicroscopy. EAS significantly attenuated S1-induced secretion of interleukin (IL)-6 in a concentration-dependent manner without reducing cell viability. EAS also markedly suppressed the S1-induced transcription of IL-6 and IL-1β. However, among the TLR4 signaling proteins, EAS did not affect the degradation of inhibitor κBα, nuclear translocation of nuclear factor-κB p65 subunit, and phosphorylation of c-Jun N-terminal kinase p54 subunit after S1 exposure. In contrast, EAS significantly suppressed S1-induced phosphorylation of p44/42 mitogen-activated protein kinase (MAPK) and Akt. Attenuation of S1-induced transcription of IL-6 and IL-1β by the MAPK kinase inhibitor U0126 was greater than that by the Akt inhibitor perifosine, and the effects were potentiated by simultaneous treatment with both inhibitors. These results suggest that EAS attenuates S1-induced IL-6 and IL-1β production by suppressing p44/42 MAPK and Akt signaling in macrophages. Therefore, EAS may be beneficial in regulating excessive inflammation in patients with COVID-19.

## 1. Introduction

Coronavirus disease 2019 (COVID-19) is an infectious disease caused by a novel type of coronavirus referred to as severe acute respiratory syndrome coronavirus 2 (SARS-CoV-2). Although 81% of infected patients develop either mild or uncomplicated illness, 14% develop pneumonia that requires hospitalization and oxygen inhalation, and 5% of patients become critically ill with respiratory failure, systemic shock, or multiple organ failure [1,2]. In particular, people with a state of low-grade systemic chronic inflammation, such as advanced aging [3,4], obesity [5,6,7], and type 2 diabetes [8,9,10], have a higher risk of death from COVID-19. Indeed, growing evidence suggests that excessive host inflammatory responses are associated with disease severity and mortality in patients [11,12].

Severe cases demonstrated markedly high levels of tumor necrosis factor-α, interleukin (IL)-6, IL-10, and the soluble form of IL-2 receptor in circulation [13], exhibiting features similar to those of cytokine storm syndromes, such as macrophage activation syndrome [12]. Macrophages produce pro-inflammatory cytokines after detecting a broad range of pathogen-associated molecular patterns (PAMPs) using pattern recognition receptors, such as toll-like receptors (TLRs). We recently reported that the SARS-CoV-2 spike protein S1 subunit (S1) strongly induces IL-6 and IL-1β production in murine and human macrophages by activating TLR4 signaling, similar to the action of pyrogenic lipopolysaccharide (LPS) [14]. Of note, recent clinical studies suggest that the IL-6 receptor antagonists tocilizumab and sarilumab and the IL-1 receptor antagonist anakinra improve the survival of patients with COVID-19 [15,16]. Since the manifestation of inflammatory disorders is influenced by an individual’s lifestyle, it is important to explore functional foods as a prophylactic approach that can suppress excessive and undesired pro-inflammatory responses in macrophages.

ETAS^®^50 is a standardized extract of *Asparagus officinalis* stem (EAS), produced by Amino Up Co., Ltd. (Sapporo, Japan). Initially, it was discovered to be a novel and unique functional food that attenuates sleep deprivation-induced stress responses and promotes sleep in mice and humans [17,18]. A subsequent study also reported that EAS intake reduced the feelings of dysphoria and fatigue, ameliorated the quality of sleep, enhanced stress-load performance, and increased salivary secretory immunoglobulin A levels in healthy adults [19]. Interestingly, a recent report showed that EAS supplementation restores cognitive functions and prevents neuronal apoptosis in the hippocampus of transgenic mice overexpressing amyloid precursor protein, which mimics Alzheimer’s disease [20]. The in vivo protective actions of EAS are suggested to be mediated by its ability to induce the expression of heat-shock protein 70 (HSP70) [17,18,20,21], and asfral [22] and asparaprolines [23] have been identified as compounds that can induce HSP70 expression at the cellular and individual levels.

Moreover, our group previously reported that EAS attenuates hydrogen peroxide-induced expression of matrix metalloproteinase 9 and pro-inflammatory mediators by suppressing phosphorylation of c-Jun N-terminal kinase (JNK) and nuclear translocation of nuclear factor-κB (NF-κB) p65 subunit, respectively, in murine skin L929 fibroblasts [24,25]. EAS also attenuated ultraviolet B-induced expression of IL-6 and IL-1β by suppressing the phosphorylation of Akt and nuclear translocation of p65, respectively, in normal human dermal fibroblasts [26,27]. These previous findings suggest that EAS has the potential to abrogate pro-inflammatory responses by inhibiting TLR4 signaling in S1-stimulated macrophages. To elucidate its potential in preventing excessive inflammation in COVID-19, we examined the effects of EAS on pro-inflammatory responses in S1-stimulated murine primary macrophages.

## 2. Results

### 2.1. EAS Attenuated S1-Induced IL-6 Secretion in a Concentration-Dependent Manner without Reducing the Cell Viability of Macrophages

To elucidate whether EAS has anti-inflammatory effects on S1-stimulated macrophages, we first examined the concentration-dependent effects of EAS treatment on S1-induced secretion of IL-6 in murine peritoneal exudate macrophages. We recently reported that S1 induced transcription and secretion of pro-inflammatory mediators in the cells in a dose-dependent manner (range: 0, 0.1, 0.5, and 1 μg/mL) and that 100 ng/mL of S1 was able to sufficiently activate TLR4 signaling [14]. Therefore, we selected 100 ng/mL as the test concentration for S1 in this study. When the cells were simultaneously treated with different concentrations (0, 0.25, 0.5, 1, or 2 mg/mL) of EAS and 100 ng/mL of S1 for 6 h, EAS significantly attenuated S1-induced secretion of IL-6 in a concentration-dependent manner (Figure 1a). To rule out that the attenuation of the S1-induced IL-6 section by EAS is due to a reduction in the number of living cells, we conducted a cell viability assay under the same experimental conditions. The cells stimulated with S1 alone showed a significant increase in cell viability, which was not influenced by EAS treatment (Figure 1b). A significant increase was also observed in the cells stimulated with LPS (Figure 1c).

### 2.2. EAS Repressed S1-Induced IL-6 and IL-1β Transcription in Macrophages

To confirm that the EAS treatment attenuates S1-induced secretion of IL-6 by transcriptional repression, we next analyzed the effects of EAS treatment on S1-induced transcription of IL-6 and IL-1β in murine peritoneal exudate macrophages. When the cells were simultaneously treated with 2 mg/mL of EAS and 100 ng/mL of S1 for 6 h, EAS dramatically repressed S1-induced transcription of IL-6 and IL-1β (Figure 2a). Moreover, EAS significantly attenuated S1-induced secretion of IL-6 even after 24 h of co-treatment (Figure 2b). Calculating the suppression rate of S1-induced IL-6 secretion by EAS (2 mg/mL) at 6 h (Figure 1a) and 24 h (Figure 2b) culture showed that extending the treatment time from 6 to 24 h significantly potentiated the attenuating effect of EAS on S1-induced secretion of IL-6 (Figure 2c). IL-1β was hardly secreted to the outside of the cells when stimulated with 100 ng/mL of S1 alone [14]. Since cleavage of the IL-1β precursor by activated caspase-1 is indispensable for the secretion of IL-1β, the cells were treated with 20 μM nigericin for the last 1 h of S1 stimulation to activate caspase-1. EAS also strongly attenuated nigericin-induced secretion of IL-1β after priming the cells with S1 stimulation for 24 h (Figure 2b).

### 2.3. EAS Suppressed S1-Induced P44/42 MAPK and Akt Phosphorylation without Affecting NF-κB Nuclear Translocation and JNK Phosphorylation in Macrophages

We recently reported that S1-induced transcription of pro-inflammatory cytokines is regulated by NF-κB and JNK signaling [14]. Therefore, we analyzed whether EAS suppresses S1-induced activation of NF-κB and JNK signaling in murine peritoneal exudate macrophages. Western blotting revealed that co-treatment of the cells with EAS (2 mg/mL) for 1 h did not affect S1 (100 ng/mL)-induced degradation of inhibitor κBα (IκBα), nuclear accumulation of NF-κB p65 subunit, and phosphorylation of JNK p54 subunit (Figure 3a). Immunofluorescence data also supported the result that EAS did not inhibit the S1-induced nuclear translocation of p65 (Figure 3b). However, instead of inhibiting the activation of NF-κB and JNK signaling, EAS significantly suppressed S1-induced phosphorylation of p44/42 MAPK and Akt after 6 h of co-treatment without interfering with basal phosphorylation levels (Figure 4a). In fact, EAS did not influence the degradation of IκBα and nuclear accumulation of the NF-κB p65 subunit by S1 exposure even after 6 h of co-treatment (Figure 4b). S1-induced JNK phosphorylation evoked transiently after 1 h of stimulation [14].

### 2.4. p44/42 MAPK Signaling Is More Involved in S1-Induced IL-6 and IL-1β Transcription in Macrophages Than Akt Signaling

To clarify whether p44/42 MAPK and Akt signaling mediate S1-induced pro-inflammatory responses, we examined the effects of the MAPK kinase inhibitor U0126 and Akt inhibitor perifosine on S1-induced transcription of IL-6 and IL-1β in murine peritoneal exudate macrophages. Co-treatment of the cells with 5 μM U0126 abolished S1-induced phosphorylation of p44/42 MAPK after 6 h (Figure 5a). At the same time, S1-induced transcription of IL-6 was significantly decreased by half, and that of IL-1β was dramatically suppressed (Figure 5b). On the other hand, co-treatment of the cells with 20 μM perifosine significantly mitigated S1-induced phosphorylation of Akt after 6 h (Figure 6a), which was accompanied by significant repression of S1-induced transcription of IL-1β, but not of IL-6 (Figure 6b). When the cells were simultaneously treated with U0126 and perifosine to mimic the condition treated with EAS, transcriptional repression of IL-6 and IL-1β demonstrated a pattern similar to that treated with U0126 alone (Figure 7a). However, calculating the suppression rate of S1-induced IL-1β and IL-6 mRNA expressions by U0126 (Figure 5b), perifosine (Figure 6b), and both (Figure 7a) showed that the inhibitory effects of U0126 on S1-induced transcription of IL-6 significantly potentiated by the addition of perifosine (Figure 7b). Although there was no statistically significant difference, a similar tendency was observed in the inhibitory effect on S1-induced transcription of IL-1β (Figure 7b).

## 3. Discussion

The pro-inflammatory cytokine IL-6 has been suggested to be involved in the aggravation of patients with COVID-19 [15,16]. In the initial experiment, we found that co-treatment with EAS attenuated S1-induced secretion of IL-6 in murine peritoneal exudate macrophages in a concentration-dependent manner after 6 h of culture. In addition, this result is not due to a reduction in the number of living cells since EAS did not influence cell viability. The viability of S1-treated cells was significantly higher than that in vehicle-treated control cells. However, primary cultured macrophages were terminally differentiated and therefore cannot undergo cell proliferation. To analyze the number of viable cells, we used a water-soluble tetrazolium salt WST-1, which was reduced to formazan dye by the activity of mitochondrial dehydrogenases in metabolically active cells. A previous study has demonstrated that the TLR4 agonist LPS facilitates mitochondrial oxidation by succinate dehydrogenase in bone marrow-derived macrophages [28]. LPS also promoted the reduction in thiazolyl blue tetrazolium bromide MTT into formazan dye in RAW264.7 macrophages [29]. We also confirmed that LPS, as well as S1, significantly increased the viability (reduction in WST-1 into formazan dye) of peritoneal exudate macrophages. Therefore, our results suggest that S1 activates the enzymes in macrophages as TLR4 agonist in a manner similar to the action of LPS and that EAS can attenuate S1-induced production of IL-6 without impairing cell survival and metabolism.

The attenuation of S1-induced IL-6 secretion is due to its transcriptional repression since EAS co-treatment dramatically repressed S1-induced transcription of IL-6 after 6 h of culture. Moreover, when the treatment time was extended from 6 to 24 h, the attenuating effect of EAS on S1-induced IL-6 secretion was clearly potentiated. These results suggest that EAS has a weak or no suppressive effect on S1-induced signal transduction in the early phase, such as NF-κB and JNK signaling, which regulates the transcription of IL-6. We recently reported that S1 induces IκBα degradation, NF-κB p65 subunit nuclear translocation, and JNK phosphorylation by interacting with TLR4 in macrophages within 1 h of culture [14]. Indeed, in this study, co-treatment with EAS did not suppress the activation of NF-κB and JNK signaling by S1 exposure. In contrast, EAS treatment suppressed NF-κB p65 nuclear translocation and JNK phosphorylation in hydrogen peroxide- or ultraviolet B-exposed skin fibroblasts [24,25,26]. The discrepancy in the action mechanisms of EAS may be due to the difference in upstream signal transduction pathways between TLR4 and other stress-triggered signaling pathways.

It has also been suggested that suppressing the action of IL-1 may be a therapeutic target for critically ill patients with COVID-19 [15]. In this study, EAS co-treatment dramatically repressed S1-induced transcription of IL-1β. The regulatory machinery for IL-1β processing and secretion substantially differs from that of most other pro-inflammatory cytokines, including IL-6. IL-1β is initially synthesized as a leaderless precursor that requires cleavage into its active form by inflammasome-activated caspase-1 [30]. The inflammasome is a multimeric protein complex consisting of the Nod-like receptor family pyrin domain containing 3 (NLRP3), apoptosis-associated speck-like protein containing a caspase recruitment domain, and pro-caspase-1, whose formation is facilitated by endogenous danger-associated molecular patterns (DAMPs), such as ATP, cholesterol crystals, urate crystals, ceramide, and nigericin [31]. Growing evidence suggests that NLRP3 inflammasome activation in macrophages and lung epithelial cells is involved in the development of pneumonia in patients with COVID-19 [32,33,34]. In this study, EAS strongly attenuated the DAMP nigericin-induced secretion of IL-1β after priming the cells with S1 stimulation for 24 h. Therefore, EAS attenuated both S1-induced IL-6 and IL-1β production by repressing their transcription in macrophages.

In addition to NF-κB and JNK signaling, the activation of p44/42 MAPK and Akt signaling was observed to regulate the transcription of pro-inflammatory cytokines in macrophages. Indeed, it has been reported that a wide variety of herbal extracts and herb-derived natural compounds exert anti-inflammatory effects on LPS-stimulated macrophage cell lines, such as RAW264.7 and U937, by suppressing either p44/42 MAPK signaling [35,36,37,38] or Akt signaling [39,40,41]. In this regard, we found that EAS co-treatment suppressed S1-induced phosphorylation of p44/42 MAPK and Akt after 6 h of culture without affecting the degradation of IκBα degradation and nuclear translocation of NF-κB p65 subunit. Moreover, S1-induced transcription of IL-6 and IL-1β was largely regulated by p44/42 MAPK signaling in murine peritoneal exudate macrophages since the transcriptional induction was significantly repressed by co-treatment with the MAPK kinase inhibitor U0126. In contrast, Akt signaling had a minor contribution to the transcriptional induction of IL-6 and IL-1β by S1 exposure compared to p44/42 MAPK signaling, as the Akt inhibitor perifosine partially repressed IL-1β transcription but did not influence IL-6 transcription. Simultaneous suppression of S1-induced phosphorylation of p44/42 MAPK and Akt augmented the repression of S1-induced IL-6 and IL-1β transcription, suggesting that EAS dramatically attenuated S1-induced IL-6 and IL-1β production by simultaneously suppressing both p44/42 MAPK and Akt signaling. However, the unknown actions of EAS should be considered because IL-6 transcription could not be dramatically repressed even when both S1-induced p44/42 and Akt phosphorylation was largely suppressed by the inhibitors. Since we analyzed the activation of NF-κB, JNK (a stress-responsive MAPK), p44/42 MAPK, and Akt, which are major signaling proteins downstream of TLR4, EAS may have complex inhibitory effects on the activity of downstream transcription factors and/or their coactivators/corepressors not only upstream signaling proteins.

More importantly, EAS did not interfere with the basal phosphorylation of p44/42 MAPK and Akt in murine primary macrophages. p44/42 MAPK is also a signal transduction protein downstream of receptor tyrosine kinases and integrins that regulate cellular motility [42]. Akt signaling acts as a master regulator for maintaining cellular proliferation, survival, and metabolism downstream of growth factor receptors [43]. Therefore, it may be extremely important for preventing impairment of macrophage functions that EAS does not indiscriminately suppress p44/42 MAPK and Akt phosphorylation. These results are also consistent with the capability of EAS to exert the attenuating effect on S1-induced IL-6 and IL-1β production in macrophages without reducing the number of living cells. Acute and subacute EAS administration had no significant side effects on food consumption, body weight, mortality, urinalysis, hematology, biochemistry, necropsy, organ weight, and histopathology in rats [44], while it can exert beneficial effects such as anti-stress, promoting sleep, and improving cognitive impairment in mice and humans [17,18,19,20]. Thus, the ability of EAS to exert various beneficial effects without side effects in vivo may be related to its failure to suppress signal transduction, which is essential for maintaining cellular functions.

A recent study showed that intratracheal administration of S1 to mice induces leukocyte infiltration, pro-inflammatory cytokine secretion, and lung tissue injury [45]. A major limitation of this study is that we have not investigated whether EAS attenuates systemic inflammation caused by S1 by the same mechanism. To date, the effects of EAS on systemic inflammation caused by PAMPs or DAMPs remain unknown. On the other hand, the in vitro and in vivo neuroprotective effects of EAS have been well demonstrated. Pre-incubation of differentiated neuronal PC-12 cells with EAS significantly restored amyloid β (Aβ)-induced reduction in cell viability, which was accompanied by reduced levels of intracellular reactive oxygen species [46]. Based on the findings obtained from this in vitro study using the cultured cell line, it was clarified that EAS intake improved the impairment of conditioned fear memory in senescence-accelerated mouse prone 8 (SAMP8) mice [47]. Moreover, EAS intake recovered cognitive performance using an active avoidance test, inhibited the expressions of Aβ precursor protein, and lowered the accumulation of Aβ in the brain of SAMP8 mice, which were accompanied by a significant increase in the number of neurons in the suprachiasmatic nucleus [48]. EAS supplementation also restored cognitive functions using a spatial learning and memory test, lowered the accumulation of Aβ, and prevented neuronal apoptosis in the hippocampus of transgenic mice overexpressing Aβ precursor protein [20]. Accumulation of Aβ in the brain has been shown to be partially triggered by inflammation [49,50], suggesting that the anti-inflammatory effect of EAS may be involved in the neuroprotective effect [48]. Therefore, based on the findings of the present study using murine primary macrophages, we would like to investigate the in vivo anti-inflammatory effect of EAS on systemic inflammation in mice administered with S1 in future studies.

## 4. Materials and Methods

### 4.1. Preparation of EAS

EAS was prepared from unused parts of asparagus in the factory of Amino Up Co., Ltd. (Sapporo, Japan) [17,18,22]. The asparagus is grown in Hokkaido, Japan, and is common in the marketplace. In brief, *Asparagus officinalis* L. stems were collected and dried and then extracted with hot water at 100 °C for 45 min. The extract was cooled to 50 °C and treated with cellulase and hemicellulase for 18 h to avoid clogging during production. After inactivation (100 °C, 20 min) of the enzymes, the extract was separated by centrifugation, concentrated in vacuo, and mixed with dextrin as a filler. The mixture was sterilized at 121 °C for 45 min and then spray-dried to produce EAS powder, consisting of 50% solid content of asparagus extract and 50% dextrin. Component analysis showed that the EAS powder comprised 78.5% carbohydrates, 12.3% proteins, 5.0% ashes, 0.7% lipids, and 3.5% moisture. The manufacturing process was conducted in accordance with suitable manufacturing practice standards for dietary supplements and ISO9001:2015 and ISO22000:2018 criteria.

### 4.2. Animal Care and Use

Adult (8–12-week-old) male C57BL/6J mice (Sankyo Labo Service, Tokyo, Japan) were housed at a temperature of 23–25 °C and humidity of 50%–60% with a fixed light/dark cycle (light, 7:00–19:00; dark, 19:00–7:00). Food and water were provided ad libitum. This study was approved by the Experimental Animal Ethics Committee in Kyorin University (no. 245, 1 April, 2021). All experiments described below were carried out following the Guiding Principles for the Care and Use of Animals approved by the Council of the Physiological Society of Japan, in accordance with the Declaration of Helsinki, 1964.

### 4.3. Preparation and Culture of Peritoneal Exudate Macrophages

Two milliliters of sterilized 4.05% thioglycollate medium Brewer modified (Becton, Dickinson and Company, Franklin Lakes, NJ, USA) was administered intraperitoneally into the mice, and the mice were housed for four days [14,51]. After the mice were euthanized by cervical dislocation, peritoneal exudate cells were harvested by sterile lavage of the peritoneal cavity with ice-cold Dulbecco’s modified Eagle’s medium (DMEM; Nacalai Tesque, Kyoto, Japan). The cells were washed once with ice-cold DMEM, resuspended in DMEM supplemented with 10% heat-inactivated fetal bovine serum (BioWest, Nuaillé, France), 100 units/mL penicillin (Nacalai Tesque), and 100 μg/mL streptomycin (Nacalai Tesque), and then cultured at 37 °C in a humidified incubator containing 5% CO_2_ for 1 h. After the nonadherent cells were removed, peritoneal exudate macrophages were used in the experiments.

### 4.4. Agents and Treatment

Different concentrations of EAS or dextrin (vehicle control) (0.25, 0.5, 1, or 2 mg/mL) were prepared by directly dissolving each agent in the complete medium, and then the supplemented medium was filter sterilized using a 0.22 μm membrane [21,24,25,26,27]. To assess its anti-inflammatory effects, the cells were co-treated with the indicated concentrations of EAS or dextrin and 100 ng/mL of SARS-CoV-2 spike recombinant protein S1 subunit (Arigo Biolaboratories, Hsinchu City, Taiwan) for 1 to 24 h. The cells were treated with 20 μM nigericin (Sigma-Aldrich, St. Louis, MO, USA) for the last 1 h of S1 stimulation to promote IL-1β processing and secretion. Cells were co-treated with 5 μM U0126 (Cell Signaling Technology, Danvers, MA, USA) or 20 μM perifosine (Cell Signaling Technology) to block S1-induced phosphorylation of p44/42 MAPK or Akt, respectively. The final concentrations of vehicles used to dissolve these agents were equivalent to the culture medium used among the experimental groups.

### 4.5. Enzyme-Linked Immunosorbent Assay (ELISA)

Cell culture supernatants were collected after centrifugation at 300× *g* for 20 min. Concentrations of IL-6 and IL-1β were measured using the Quantikine Mouse IL-6 ELISA Kit (R&D Systems, Minneapolis, MN, USA) and Quantikine Mouse IL-1β ELISA Kit (R&D Systems), as described previously [14,51]. Since the detection limits of IL-6 and IL-1β for ELISA are 7.8–500 and 12.5–800 pg/mL, respectively, the supernatants were diluted 10 times before measurements.

### 4.6. Cell Viability Assay

Cell viability was measured using the Cell Counting Kit-8 (FUJIFILM Wako Pure Chemical, Osaka, Japan) [52]. After cell culture supernatants were collected, fresh complete medium supplemented with 10% Cell Counting Kit-8 reagent was added to each well, and incubation was continued for 1 h. After the reaction was stopped by adding 1% sodium dodecyl sulfate, the absorbance of the medium in each well was analyzed using a multi-mode microplate reader FilterMax F5 (Molecular Devices, San Jose, CA, USA) at a wavelength of 450 nm. Cell viability was calculated with the viability of vehicle-treated control cells as 100%.

### 4.7. Reverse Transcription and Real-Time Polymerase Chain Reaction (PCR)

Total cellular RNA was extracted using RNAiso Plus reagent (TaKaRa Bio, Shiga, Japan). One microgram of total cellular RNA was converted to single-stranded cDNA using the PrimeScript 1st Strand cDNA Synthesis Kit (Takara Bio). The cDNA (1 μL) was amplified using Premix Ex Taq (Probe qPCR) (TaKaRa Bio) in a 7500 Real-Time PCR System (Thermo Fisher Scientific, Waltham, MA, USA). The PCR conditions were as follows: 50 °C for 2 min and 95 °C for 15 s, followed by 40 cycles of 95 °C for 15 s and 60 °C for 1 min. The fluorescent probes and primers used are listed in Table 1. The mRNA expression levels of target genes were calculated as the ratio of their values to that of 18S rRNA as an internal control.

### 4.8. Preparation of Nuclear Extracts

Nuclear proteins were prepared as previously described [14,25,26]. The cells were extracted in lysis buffer containing 10 mM HEPES–KOH (pH 7.8), 10 mM KCl, 2 mM MgCl_2_, 0.1 mM ethylenediaminetetraacetic acid (EDTA), and 0.1% Nonidet P-40 supplemented with protease and phosphatase inhibitor cocktails (Nacalai Tesque). After low-speed centrifugation (200× *g*) at 4 °C for 5 min, sediments containing the nuclei were resuspended in wash buffer containing 250 mM sucrose, 10 mM HEPES–KOH (pH 7.8), 10 mM KCl, 2 mM MgCl_2_, and 0.1 mM EDTA. The suspension was centrifuged at low speed (200× *g*) at 4 °C for 5 min. Then, the sediments were resuspended in nuclear extraction buffer containing 50 mM HEPES–KOH (pH 7.8), 420 mM KCl, 5 mM MgCl_2_, 0.1 mM EDTA, and 20% glycerol, and rotated at 4 °C for 30 min. After high-speed centrifugation (13,000× *g*) at 4 °C for 15 min, the supernatants were used as a source of nuclear proteins. The concentration of nuclear proteins was determined using the Protein Assay BCA kit (Nacalai Tesque).

### 4.9. Western Blotting

Whole or nuclear proteins (10 μg) were separated through electrophoresis on a sodium dodecyl sulfate-polyacrylamide gel and then transferred onto an Immobilon-P membrane (Merck Millipore, Burlington, MA, USA). After blocking with 5% bovine serum albumin, each membrane was probed with primary antibodies for 1 h (Table 2). Secondary antibodies conjugated with horseradish peroxidase (Jackson ImmunoResearch Laboratories, West Grove, PA, USA) were applied at a 1:20,000 dilution for 30 min. The membrane was incubated with Western BLoT Chemiluminescence HRP Substrate (TaKaRa Bio), and membrane image was captured using a chemiluminescent imaging system LuminoGraph I (ATTO, Tokyo, Japan). The density of each protein band was quantified using ImageJ software (National Institutes of Health, Bethesda, MD, USA). Glyceraldehyde-3-phosphate dehydrogenase (GAPDH) and Yin Yang 1 (YY1) were used as loading controls for whole and nuclear proteins, respectively. The phosphorylation levels of the target proteins were calculated as the ratio of their values to those of the corresponding total protein as a loading control.

### 4.10. Fluorescence Immunomicroscopy

The cells were fixed with 4% paraformaldehyde for 15 min and then permeabilized with methanol at −20 °C for 10 min. After blocking with 1% bovine serum albumin, the primary antibody against p65 (8242S; Cell Signaling Technology) was applied at a 1:400 dilution for 1 h. Then, the secondary antibody conjugated with Alexa Fluor 568 (Abcam, Cambridge, UK) was applied at a 1:1000 dilution with 3 μM Nuclear Green DCS1 (Abcam) for 30 min. After mounting on glass slides, the subcellular localization of p65 and the nucleus was visualized with FL2 (orange-red) and FL1 (green) detectors, respectively, using Nikon BioStation IM (NIKON, Tokyo, Japan) [14,26]. The brightness of green and orange-red fluorescence was quantified using ImageJ software (National Institutes of Health, Bethesda, MD, USA). The amount of p65 in the nucleus was calculated as the ratio of the brightness of orange-red fluorescence to that of the corresponding green fluorescence.

### 4.11. Statistical Analysis

Experimental data are presented as mean ± standard error of the mean (SEM). Differences between the two groups were assessed using the Student’s *t*-test. Comparisons among at least three groups were tested using one-way analysis of variance (ANOVA), and post-hoc comparisons to determine significant differences between the two groups were performed using the Bonferroni test. Differences were considered statistically significant at *p* < 0.05.

## 5. Conclusions

Excessive host inflammatory responses are associated with disease severity and mortality in patients with COVID-19. In particular, suppressing IL-6 and IL-1 production and blocking their receptors and actions are currently attracting attention as therapeutic targets for severe COVID-19. Growing evidence also suggests that the SARS-CoV-2 spike protein contributes to the development of macrophage activation syndrome by activating TLR4 signaling. In this study, EAS attenuated S1-induced IL-6 and IL-1β production by suppressing p44/42 MAPK and Akt phosphorylation in murine primary macrophages. Therefore, this readily available, inexpensive, and eco-friendly functional food may be a useful component in prophylactic strategies for regulating excessive inflammation in patients with COVID-19.

## Figures and Tables

**Figure 1 molecules-26-06189-f001:**
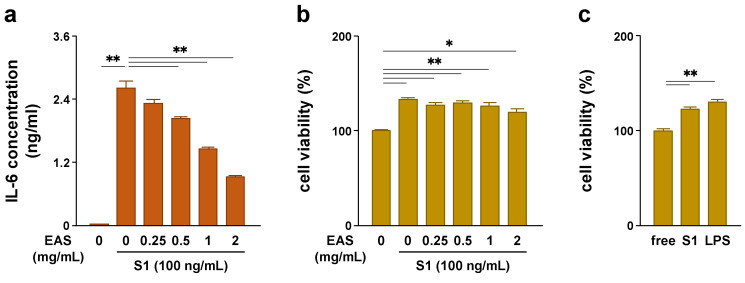
Effect of EAS on S1-induced secretion of IL-6 and viability of murine peritoneal exudate macrophages. (**a,b**) The cells were co-treated with 0, 0.25, 0.5, 1, or 2 mg/mL of EAS and 100 ng/mL of S1 for 6 h. (**a**) IL-6 concentrations in culture supernatants were analyzed using ELISA. (**b**) Cell viability was analyzed using the Cell Counting Kit-8. (**c**) The cells were treated with 100 ng/mL of LPS or S1 for 6 h. Cell viability was analyzed using the Cell Counting Kit-8. Mean ± SEM (*n* = 3). * *p* < 0.05, ** *p* < 0.01, using one-way ANOVA and Bonferroni test.

**Figure 2 molecules-26-06189-f002:**
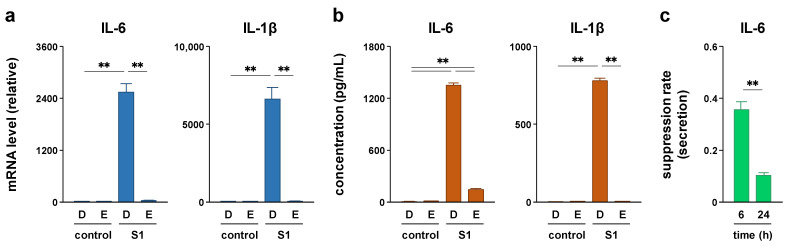
Effect of EAS on S1-induced transcription of IL-6 and IL-1β in murine peritoneal exudate macrophages. The cells were co-treated with 2 mg/mL of EAS (E) or dextrin (D; vehicle control) and 100 ng/mL of S1 for 6 h (**a**) and 24 h (**b**). (**a**) IL-6 (left) and IL-1β (right) mRNA levels were analyzed using real-time PCR. (**b**) IL-6 (left) and IL-1β (right) concentrations in culture supernatants were analyzed using ELISA. The cells were treated with 20 μM nigericin for the last 1 h of S1 stimulation to promote IL-1β processing and secretion ((**b**) right). (**c**) The suppression rate of S1-induced IL-6 secretion by EAS (2 mg/mL) at 6 h (Figure 1a) and 24 h (Figure 2b) culture was calculated by dividing the EAS-treated values by the corresponding EAS-free (dextrin) values. Mean ± SEM (*n* = 3). ** *p* < 0.01, using one-way ANOVA and Bonferroni test (**a,b**) and Student’s *t*-test (**c**).

**Figure 3 molecules-26-06189-f003:**
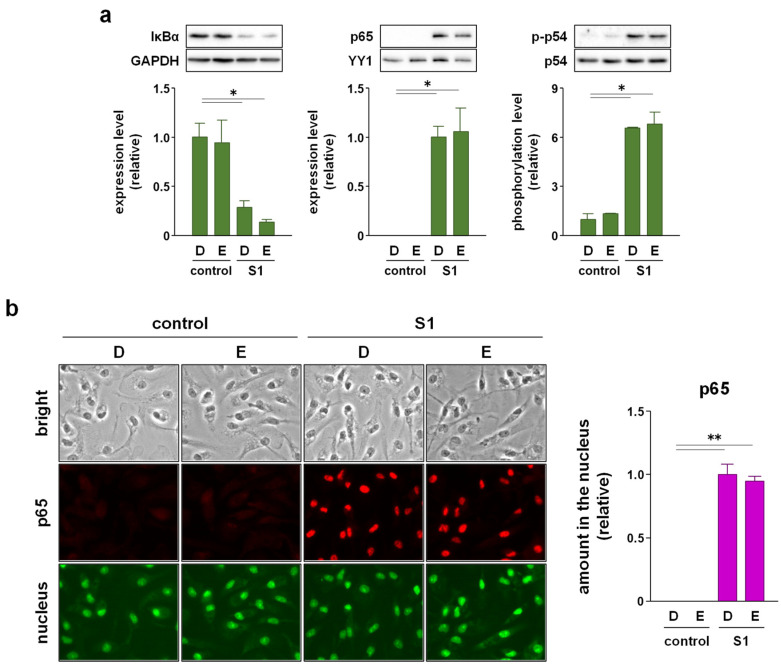
Effect of EAS on S1-induced activation of the NF-κB and JNK signaling in murine peritoneal exudate macrophages. The cells were co-treated with 2 mg/mL of EAS (E) or dextrin (D; vehicle control) and 100 ng/mL of S1 for 1 h. (**a**) Total amount of IκBα (left), nuclear amount of NF-κB p65 subunit (middle), and phosphorylation level of JNK p54 subunit (right) were analyzed using western blotting. Mean ± SEM (*n* = 3). * *p* < 0.05, using one-way ANOVA and Bonferroni test. (**b**) Subcellular localization of p65 was visualized using fluorescence immunomicroscopy. Mean ± SEM (*n* = 10). ** *p* < 0.01, using one-way ANOVA and Bonferroni test.

**Figure 4 molecules-26-06189-f004:**
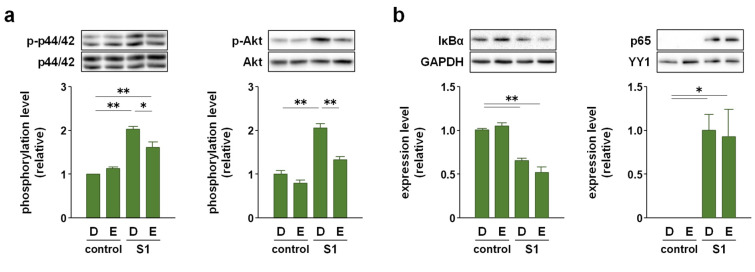
Effect of EAS on S1-induced phosphorylation of p44/42 MAPK and Akt in murine peritoneal exudate macrophages. The cells were co-treated with 2 mg/mL of EAS (E) or dextrin (D; vehicle control), and 100 ng/mL of S1 for 6 h. (**a**) Phosphorylation levels of p44/42 MAPK (left) and Akt (right) were analyzed using western blotting. (**b**) Total amount of IκBα (left) and nuclear amount of NF-κB p65 subunit (right) were analyzed using western blotting. Mean ± SEM (*n* = 3). * *p* < 0.05, ** *p* < 0.01, using one-way ANOVA and Bonferroni test.

**Figure 5 molecules-26-06189-f005:**
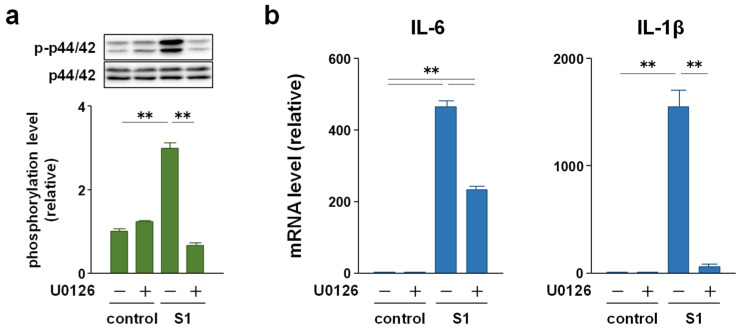
Effect of the MAPK inhibitor U0126 on S1-induced transcription of IL-6 and IL-1β in murine peritoneal exudate macrophages. The cells were co-treated with 5 μM U0126 or dimethyl sulfoxide alone (vehicle control) and 100 ng/mL of S1 for 6 h. (**a**) Phosphorylation level of p44/42 MAPK was analyzed using western blotting. (**b**) IL-6 (left) and IL-1β (right) mRNA levels were analyzed using real-time PCR. Mean ± SEM (*n* = 3). ** *p* < 0.01, using one-way ANOVA and Bonferroni test.

**Figure 6 molecules-26-06189-f006:**
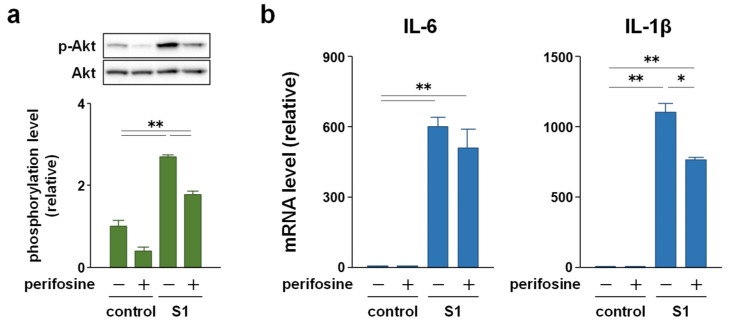
Effect of the Akt inhibitor perifosine on S1-induced transcription of IL-6 and IL-1β in murine peritoneal exudate macrophages. The cells were co-treated with 20 μM perifosine or sterile water alone (vehicle control) and 100 ng/mL of S1 for 6 h. (**a**) Phosphorylation level of Akt was analyzed using western blotting. (**b**) IL-6 (left) and IL-1β (right) mRNA levels were analyzed using real-time PCR. Mean ± SEM (*n* = 3). * *p* < 0.05, ** *p* < 0.01, using one-way ANOVA and Bonferroni test.

**Figure 7 molecules-26-06189-f007:**
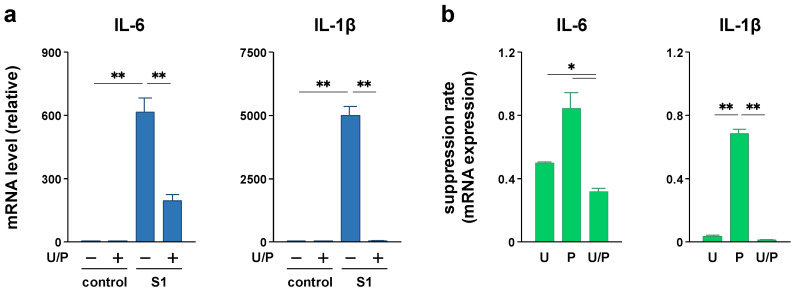
Effect of simultaneous treatment with U0126 and perifosine on S1-induced transcription of IL-6 and IL-1β in murine peritoneal exudate macrophages. (**a**) The cells were co-treated with 5 μM U0126 (U), 20 μM perifosine (P), and 100 ng/mL of S1 for 6 h. The concentration of vehicles is the same across all samples. IL-6 (left) and IL-1β (right) mRNA levels were analyzed using real-time PCR. (**b**) The suppression rate of S1-induced IL-1β and IL-6 mRNA expressions by U0126 (Figure 5b), perifosine (Figure 6b), and both (Figure 7a) were calculated by dividing each inhibitor-treated value by the corresponding inhibitor-free values. Mean ± SEM (*n* = 3). * *p* < 0.05, ** *p* < 0.01, using one-way ANOVA and Bonferroni test.

**Table 1 molecules-26-06189-t001:** Fluorescent probes and primers used in this study.

Species	Gene	Probe	Forward Primer Sequence	Reverse Primer Sequence
Mouse	*Il6*	#6 (Roche)	5’-GAT GGA TGC TAC CAA ACT GGA-3’	5’-CCA GGT AGC TAT GGT ACT CCA GAA-3’
Mouse	*Il1b*	#78 (Roche)	5’-TGT AAT GAA AGA CGG CAC ACC-3’	5’-TCT TCT TTG GGT ATT GCT TGG-3’
Mouse	*Rn18s*	#55 (Roche)	5’-GGA GAA AAT CTG GCA CCA CAC CTT-3’	5’-CCT TAA TGT CAC GCA CGA TTT CCC-3’

Roche: Roche Life Science, Indianapolis, IN, USA.

**Table 2 molecules-26-06189-t002:** Primary antibodies used in this study.

Protein	m.w.	Cat. No.	Manufacturer	Dilution	Protein	m.w.	Cat. No.	Manufacturer	Dilution
IκBα	39 kDa	4814S	CST	1/1000	GAPDH	37 kDa	5174S	CST	1/2000
p65	65 kDa	8242S	CST	1/1000	YY1	68 kDa	ab109237	Abcam	1/2000
p-JNK	54/46 kDa	4668S	CST	1/1000	JNK	54/46 kDa	9258S	CST	1/1000
p-p44/42	44/42 kDa	4370P	CST	1/1000	p44/42	44/42 kDa	4695P	CST	1/1000
p-Akt	60 kDa	4060S	CST	1/1000	Akt	60 kDa	4691S	CST	1/1000

CST: Cell Signaling Technology, Danvers, MA, USA.

## Data Availability

The data presented in this study are available on request from the corresponding author.

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
