# Peer review of "Standardized Extract of Asparagus officinalis Stem Attenuates SARS-CoV-2 Spike Protein-Induced IL-6 and IL-1β Production by Suppressing p44/42 MAPK and Akt Phosphorylation in Murine Primary Macrophages"

_molecules, 2021, doi:10.3390/molecules26206189_

Round 1

Reviewer 1 Report

The authors present the use of EAS as an anti-inflammatory compound against the spike region of COVDI-19. They show that EAS is able to inhibit macrophage activation by S1.

The paper is well written, experiments were properly executed and presented clearly.  However, I do not find the findings to be of high significance. The data generated were done all in vitro, it is not known if EAS when provided in vivo can act in the same manner. Perhaps this can be included in the discussion if there is the available information on its anti-inflammatory properties in vivo.  The authors have touched a little bit that EAS was trialled in humans and mice.

The authors claim that MAPK and Akt signalling is additive, but Figure 7, is similar to U0126 inhibition alone. While the scales for IL6 expression is similar for Figures 5-7, IL1beta expression varies quite highly. The interpretation that the effect is additive is not convincing.  Scale bars if kept consistent will show this. EAS was able to completely inhibit IL-6 expression, and should also be mentioned as this suggest other pathways may also be at play.

Can the authors also explain why nigericin is required for IL-1beta release?

Author Response

List of corrections and replies to Reviewer 1

Thank you very much for your valuable and adequate comments and suggestions that help us to improve the quality of our study. In particular, we have added discussion about the potential in vivo anti-inflammatory effects of EAS, corrected the interpretation of the data based on the data re-analysis, and revised the manuscript according to your comments and suggestions. List of corrections, additions, or responses to you is as follows:

1) Response to the 1st comments and suggestions: The paper is well written, experiments were properly executed and presented clearly. However, I do not find the findings to be of high significance. The data generated were done all in vitro, it is not known if EAS when provided in vivo can act in the same manner. Perhaps this can be included in the discussion if there is the available information on its anti-inflammatory properties in vivo. The authors have touched a little bit that EAS was trialled in humans and mice.

Thank you very much for your valuable comments, advice, and suggestion. As you pointed out, a major limitation of this study is that we have not investigated whether EAS attenuates systemic inflammation caused by S1 by the same mechanism. To date, the effects of EAS intake on systemic inflammation caused by PAMPs or DAMPs remain unknown. On the other hand, the in vitro and in vivo neuroprotective effects of EAS have been well demonstrated, and the preventive effects on the accumulation of amyloid-β in the brain are suggested to be partially mediated by the anti-inflammatory effects of EAS. Therefore, we have added the limitation of the present study and discussed potential in vivo anti-inflammatory effects of EAS (please see red lines 413-435 in the revised manuscript) with citing several additional literatures (please see red lines 601-618 in the revised manuscript). In this regard, “pathogen-associated molecular patterns” was abbreviated to “PAMPs” (please see red line 53 in the revised manuscript).

We consider that clarifying the biological and pharmacological effects of plant extracts and the mechanisms at the cellular level is important as a scientific procedure. Ethical reviews of animal experiments in the medical schools have become stricter in recent years. Indeed, recent studies have shown that S1 administration to mice evokes severe tissue and systemic inflammation. Therefore, obtaining approval from the scientific community for the beneficial effects of EAS in vitro and for the validity of our research methods can facilitate the ethical application of future animal experiments. We greatly appreciate your cooperation.

2) Response to the 2nd comments and suggestions: The authors claim that MAPK and Akt signalling is additive, but Figure 7, is similar to U0126 inhibition alone. While the scales for IL-6 expression is similar for Figures 5-7, IL-1beta expression varies quite highly. The interpretation that the effect is additive is not convincing. Scale bars if kept consistent will show this. EAS was able to completely inhibit IL-6 expression, and should also be mentioned as this suggest other pathways may also be at play.

Thank you very much for your helpful comments, advice, and suggestion. Since the mRNA levels of cytokines are very low when the cells are unstimulated, the error is also large especially with IL-1β. This causes the scale of relative expression levels of mRNA to vary greatly for each experiment. Therefore, the suppression rate of S1-induced IL-6 and IL-1β mRNA expressions by the inhibitors was calculated and shown as a single figure (Figure 7b) so that we have made it possible for readers to compare at a glance how many times S1-induced transcription of IL-6 and IL-1β mRNA was suppressed by U0126, perifosine, and both. From this data re-analysis, because we found that “additively” was not a proper description as you pointed out, the interpretation was corrected. Moreover, the existence of unknown pathways of EAS actions were also mentioned. Ultimately, we have revised the manuscript as follows:

1) Results: The title of subheading 3.4 was corrected (please see red lines 288-289 in the revised manuscript). In addition, the interpretation of the data in Figure 7a was corrected (please see red lines 299-301 in the revised manuscript), and details of the additional data re-analysis (Figure 7b) were added (please see red lines 301-306 in the revised manuscript).

2) Figure: Figure 7b (please see line 317 in the revised manuscript) and its legend (please see red lines 321-323 in the revised manuscript) was added.

3) Discussion: The existence of unknown pathways of EAS actions were mentioned (please see red lines 390-396 in the revised manuscript).

3) Response to the 3rd comments and suggestions: Can the authors also explain why nigericin is required for IL-1beta release?

Thank you very much for your question. We recently reported that IL-1β was hardly secreted to the outside of the cells when stimulated with 100 ng/mL of S1 alone (Shirato, K. & Kizaki, T. Heliyon 2021, 7, e06187). Since cleavage of the IL-1β precursor by activated caspase-1 is indispensable for sufficient secretion of IL-1β, the cells were treated with 20 μM nigericin for the last 1 h of S1 stimulation to activate caspase-1 in this study. We have added this background why treatment with a DAMP nigericin was conducted into the Results section (please see red lines 245-248 in the revised manuscript). In the Discussion section, we also added nigericin into the DAMP list (please see red line 367 in the revised manuscript). In addition, detailed mechanism of IL-1β maturation and secretion and the involvement of DAMPs-induced caspase-1 activation and IL-1β secretion in pathogenesis of COVID-19 are discussed on line 357-372.

Reviewer 2 Report

The manuscript entitled "Standardized Extract of Asparagus officinalis Stem Attenuates SARS-CoV-2 Spike Protein-Induced IL-6 and IL-1β Production by Suppressing p44/42 MAPK and Akt Phosphorylation in Murine Primary Macrophages" seems an interesting piece of work. However, there are many flaws in the manuscript, which needs to be addressed, before being accepted for publications.

  1. The authors selected 100 ng/mL of SARS-CoV-2 spike recombinant protein S1 subunit to stimulate murine primary macrophages in this work, is there any specific reason for selecting this concentration (100 ng/mL) for stimulation? If the authors provide the information on the concentration gradient experiment of S1, it could be more informative.
  2. In Results section 3.2, the authors mentioned that “Moreover, extending the treatment time from 6 h to 24 h potentiated the attenuating effect of EAS on S1-induced secretion of IL-6 (Figure 1a,2b)”, which seemed to be a little confused. If authors show the comparing results of these two time points in one figure, it would help the reader to understand.
  3. The titles in the Figures 2a, 5b, 6b, and 7 were “Il6 and Il1b”, but in Figure 2b were “IL-6” and “IL-1β”, please correct them as the same pattern.
  4. The results of immunofluorescence data in Results section 3.3 (Figure 3b) need to perform the statistical analysis, please add it.
  5. In Results section 3.3, the authors mentioned that the western blotting assays of IκBα, p65 and p-p54 were performed on the cells co-treated with EAS and S1 for 1h, but the test of p-p44/42 and p-Akt were performed on the cells co-treated with EAS and S1 for 6h. The experiments were not conducted at the same time point, but the authors compared these proteins expression at two time points and draw a conclusion that “EAS suppressed S1-induced p44/42 MAPK and Akt phosphorylation without affecting NF-κB nuclear translocation and JNK phosphorylation in macrophages”, which need clarification and addition of more details. Please make it clear.
  6. In Discussion section (the first paragraph), the authors stated that “Therefore, our results suggest that S1 activates the enzymes in macrophages as TLR4 agonist in a manner similar to the action of LPS”, there are no experimental results in this study to prove this conclusion, please clarify it clearly.

Author Response

List of corrections and replies to Reviewer 2

Thank you very much for your valuable and adequate comments and suggestions that help us to improve the quality of our study. In particular, we have conducted additional experiments, modified the interpretation of the data based on the data re-analysis, and revised the manuscript according to your comments and suggestions. List of corrections, additions, or responses to you is as follows:

1) Response to the 1st comments and suggestions: The authors selected 100 ng/mL of SARS-CoV-2 spike recombinant protein S1 subunit to stimulate murine primary macrophages in this work, is there any specific reason for selecting this concentration (100 ng/mL) for stimulation? If the authors provide the information on the concentration gradient experiment of S1, it could be more informative.

Thank you very much for your helpful comments, advice, and suggestion. As you suggested, we recently reported that S1 induced transcription and secretion of pro-inflammatory mediators in murine peritoneal exudate macrophages in a dose-dependent manner (range: 0, 0.1, 0.5, and 1 μg/mL) and that 100 ng/mL of S1 was able to sufficiently activate TLR4 signaling (Shirato, K. & Kizaki, T. Heliyon 2021, 7, e06187). We added into the Results section that we selected the concentration based on these previous findings (please see red lines 217-221 in the revised manuscript).

2) Response to the 2nd comments and suggestions: In Results section 3.2, the authors mentioned that “Moreover, extending the treatment time from 6 h to 24 h potentiated the attenuating effect of EAS on S1-induced secretion of IL-6 (Figure 1a,2b)”, which seemed to be a little confused. If authors show the comparing results of these two time points in one figure, it would help the reader to understand.

Thank you very much for your helpful comments, advice, and suggestion. According to your advice, to avoid confusion for readers, we have added Figure 2c, which compares the results at two time points. Instead of comparing the raw data, the suppression rate of S1-induced IL-6 secretion by EAS (2 mg/mL) at each time point was calculated and the statistical analysis was performed. Therefore, we have revised the manuscript as follows:

1) Materials and Methods: The procedure for statistical analysis was added (please see red lines 206-207 in the revised manuscript).

2) Results: Details of the additional data re-analysis were added (please see red lines 240-245 in the revised manuscript).

3) Figure: Figure 2c (please see line 252) and its legend (please see red lines 257-259 in the revised manuscript) was added.

3) Response to the 3rd comments and suggestions: The titles in the Figures 2a, 5b, 6b, and 7 were “Il6 and Il1b”, but in Figure 2b were “IL-6” and “IL-1β”, please correct them as the same pattern.

Thank you very much for your correction. We have corrected the descriptions and integrated into “IL-6” and “IL-1β”.

4) Response to the 4th comments and suggestions: The results of immunofluorescence data in Results section 3.3 (Figure 3b) need to perform the statistical analysis, please add it.

Thank you very much for your valuable comments and advice. We have quantified the immunofluorescence data and the statistical analysis has been performed. We have revised the manuscript as follows:

1) Materials and methods: Procedures for the quantification were added (please see red lines 201-204 in the revised manuscript).

2) Figure: The quantification and statistics were added into Figure 3b (Please see line 276). Details of the statistics were added into Figure legend (please see red line 281 in the revised manuscript).

5) Response to the 5th comments and suggestions: In Results section 3.3, the authors mentioned that the western blotting assays of IκBα, p65 and p-p54 were performed on the cells co-treated with EAS and S1 for 1h, but the test of p-p44/42 and p-Akt were performed on the cells co-treated with EAS and S1 for 6h. The experiments were not conducted at the same time point, but the authors compared these proteins expression at two time points and draw a conclusion that “EAS suppressed S1-induced p44/42 MAPK and Akt phosphorylation without affecting NF-κB nuclear translocation and JNK phosphorylation in macrophages”, which need clarification and addition of more details. Please make it clear.

Thank you very much for your valuable comments and suggestion. According to your comments and suggestion, we have conducted additional experiments, and found that EAS did not influence degradation of IκBα and nuclear accumulation of NF-κB p65 subunit by S1 exposure even after 6 h of co-treatment (Figure 4b). Therefore, we have revised the manuscript as follows:

1) Results: Details of the results were added (please see red lines 272-275 in the revised manuscript).

2) Figure: Figure 4b (please see line 282) and its legend were added (please see red line 285-286 in the revised manuscript).

3) Discussion: The results were added to support our conclusion (please see red lines 379-380 in the revised manuscript).

6) Response to the 6th comments and suggestions: In Discussion section (the first paragraph), the authors stated that “Therefore, our results suggest that S1 activates the enzymes in macrophages as TLR4 agonist in a manner similar to the action of LPS”, there are no experimental results in this study to prove this conclusion, please clarify it clearly.

Thank you very much for your sharp point. According to you point, we have conducted additional experiments, and found that a significant increase in the viability (reduction of WST-1 into formazan dye) was observed in the cells stimulated with LPS as well as S1. This positive control data has been added in the revised manuscript to support our speculation. Therefore, we have revised the manuscript as follows:

1) Results: The results were added (please see red line 228 in the revised manuscript).

2) Figure: Figure 1c (please see line 229) and its legend (please see red lines 232-233 in the revised manuscript).

3) Discussion: The results were added to support our speculation (please see red lines 338-340 in the revised manuscript).

Round 2

Reviewer 1 Report

I am happy with the changes the authors have made.